# The global *viralization* of policies to contain the spreading of the COVID-19 pandemic: Analyses of school closures and first reported cases

José Ignacio Nazif-Muñoz[1,2]*, Sebastián Peña[3,4], Youssef Oulhote[2,5]

**1** Faculty of Medicine and Health Sciences, Université de Sherbrooke, Longueuil, Canada, **2** Department of Environmental Health, T. H. Chan School of Public Health, Harvard University, Boston, Massachusetts, United States of America, **3** Department of Public Health Solutions, Finnish Institute for Health and Welfare (THL), Helsinki, Finland, **4** Faculty of Medicine, Universidad Diego Portales, Santiago, Chile, **5** Department of Biostatistics and Epidemiology, School of Public Health and Health Sciences, University of Massachusetts at Amherst, Amherst, Massachusetts, United States of America

* jjose.ignacio.nazif-munoz@usherbrooke.ca

**Data Availability Statement:** All relevant data are within the manuscript and its Supporting information files.

## Abstract

### Background

On January 30[th] 2020, the World Health Organization (WHO) declared a international health emergency due to the unprecedented phenomenon of COVID-19. After this declaration countries swiftly implemented a variety of health policies. In this work we examine how rapid countries responded to this pandemic using two events: the day in which the first case of COVID-19 was reported, and first day in which countries used school closure as one of the measures to avoid outbreaks. We also assessed how countries' health systems, globalization, economic development, political systems, and economic integration to China, Republic of Korea and Italy increased the speed of adoption.

### Methods

We compiled information from multiple sources, from December 31[st] 2019 to June 1[st] 2020, to trace when 172 countries reported their first COVID-19 case and implemented school closure to contain outbreaks. We applied cross-national Weibull survival analysis to evaluate the global speed of detection of first COVID-19 reported cases and school closure.

### Results

Ten days after WHO declared COVID-19 to be an international emergency, relative to seven days from this declaration, countries were 28 (95% CI: 12–77) times more likely to report first COVID-19 cases and 42 (95% CI: 22–90) times more likely to close schools. One standard deviation increase in the epidemic security index rises the rate of report first cases by 37% (Hazard Ratio (HR) 1.37 (95% CI: 1.09–1.72) and delays the adoption for school closures by 36% (HR 0.64 (95% CI:0.50–0.82). One standard deviation increase in the

**Funding:** JINM received funds from the Fonds de recherche du Québec – Santé. Junior 1 http://www.frqs.gouv.qc.ca/.

**Competing interests:** The authors declare that no competing interests exist regarding this manuscript.

globalization index augments the adoption for school closures by 74% (HR 1.74 (95% CI:1.34–2.24).

## Conclusion

After the WHO declared a global emergency, countries were unprecedently acting very rapidly. While countries more globally integrated were swifter in closing schools, countries with better designed health systems to tackle epidemics were slower in adopting it. More studies are needed to assess how the speed of school closures and other policies will affect the development of the pandemic.

## Introduction

In the past thirty years a strong tendency towards isomorphism has been identified across nation-states [1]. That is countries are likely to have similar public policy responses when faced with challenges that occur at the national or international levels. In short, we have observed in this period a trend of global convergence of multiple policies such as democracy, terrorist laws, privatization, human rights among others [2–9]. With the current exposure to COVID-19 pandemic, countries rapidly realigned the allocation of public resources after China informed the World Health Organization's (WHO) authorities (December 31st 2019) [10, 11], and Italy issued a state emergency decree the same date when WHO declared a global health emergency (January 30th 2020) [12]. As such this phenomenon provides a unique opportunity to revisit the thesis of global convergence with a rather exceptionally short time span of five months.

Simultaneously, the debate over what measures should have been implemented to prevent deaths associated with vulnerable populations or at least decrease the lethality of this virus has been fierce [12]. While some measures seemed to be more effective than others in Singapore [13] and Republic of Korea—including aggressive technological tracing, massive testing, and isolation of cases and extensive quarantining of contacts—many concerns were raised since some public health recommendations directly affected liberties and economies, and thus the overall functioning of countries, regions and the world [14, 15]. Even though countries' decisions to tackle the pandemic is suggesting a strong case of global convergence yet little is known of what makes countries to more rapidly adopt such policies. Indeed, the presence of uneven public health resources across countries, different levels of integration with the world, China, Republic of Korea and Italy, respect for liberties, may have all constrained differently countries' swiftness to rapid response. In short, the COVID-19 pandemic offers an exclusive opportunity to explore how countries transit this public health emergency by rapidly introducing different policies.

In this work, we examine the speed of isomorphism using two events associated with the COVID-19 pandemic: 1) the day in which the first case of COVID-19 was reported in each country, and 2) the first day in which countries nationally used school closure as one of the measures to reduce the spread of this virus. We chose school closing since its implementation informs a sense of urgency under uncertain conditions while little was known about transmission in children [16].

Since countries are not similarly prepared, we hypothesise that countries with greater global political and economic integration [17], and neither equally distant to China, South Korea nor Italy, will respond at different speeds to these two events. We advance two groups of

**Table 1. Hypotheses for first reported case of COVID-19 and school closures.**

| Events | Hypotheses |
|---|---|
| 1. *First reported case of COVID-19* | 1.1 After the WHO declared a global health emergency, countries will be more likely to report first cases of COVID-19. |
| | 1.2 Countries with health systems designed to respond and mitigate more rapidly the spread of an epidemic, higher gross development product (GDP) per capita, more populated, more globally integrated and with tighter economic ties to China, Republic of Korea or Italy will be more rapid in reporting to the first detected case. |
| 2. *School closures* | 2.1 Countries more globally integrated will be more exposed to the influence of the WHO recommendations to tackle the pandemic and therefore more rapidly to adopt school closure. |
| | 2.2 Countries with health systems designed to respond and mitigate more rapidly the spread of an epidemic will delay the implementation of school closures since they have better knowledge to prevent a stringent measure such as school closures |
| | 2.3 Countries with higher GDP will delay the implementation of school closures since this measure has a direct impact in the economy. |
| | 2.4 Less democratic countries will be swifter in implementing this measure since a vertical response of this nature implies a direct limitation of freedom of assembly, which in these countries may not be regarded as a fundamental right. |
| | 2.5 Countries more economically integrated with China, Republic of Korea and Italy will be more rapidly closing schools since closeness to these countries will raise higher public health concerns to stop the spreading of the COVID-19. |

hypotheses, one for the detection of the first case of COVID-19, and another for school closures. In Table 1 we formalize each hypothesis for both events.

## Methods and data

To test the speed of the global convergence thesis in the context of the COVID-19 pandemic and what makes countries more rapidly adopt policies designed to detect and stop its contagion, we compiled and analyzed data on 172 countries between December 31st 2019 and June 1st 2020 from multiple sources and applied survival analysis. Each source to carry out the analyses is described below and open repositories.

### i) Outcomes

We analyzed two events: i) date in which the first case of COVID-19 was reported; and ii) date in which schools were closed at the national level. (In France the first date of school closure occurred at the provincial level in March 7th 2020, however the national decision to implement this measure took place in March 16th [18]; this date was used to code when the adoption occurred in this country). Countries in which a national decision was not taken before June 1st 2020 were regarded as not having implemented this measure (i.e. censored). To check for the robustness of this design, we also carried out analyses with countries in which national decision was not taken before this date and used the last date in which a state or province had reported closure and results were consistent. Information to detect dates in which first cases of COVID-19 were reported, and school closure was carried out was gathered from UNESCO [18] as well as from governments' websites and national and international newspapers (S1 File contains all sources per country). To verify information on when first cases of COVID-19 were reported per country, we also used the open depository worldometer [19].

## ii) Determinants of early response

We use the following variables to explore what makes countries adopt more rapidly or more slowly the two events.

**Epidemic security index.**   We expect this variable to increase the hazard rate of reporting the first case since these countries would have more capacity to detect the presence of the virus in the population. We expect this variable to be associated with a less rapid adoption of school closure since other measures are likely to be assessed before setting in place a very restrictive measure in the population. These data have been obtained from the Global Health Security Index [20]. We used data corresponding to the dimension "rapidly responding to and mitigating the spread of an epidemic" which gathers national information on: Emergency preparedness and response planning; exercising response plans; emergency response operation; linking public health and security authorities; risk communication; access to communications infrastructure; and trade and travel restrictions. To facilitate interpretation of this index we transformed values to z-scores.

**Globalization index.**   This a measure of globalization and global influence, considering social, political and economic dimensions. We expect regimes with higher levels of social, political, and economic integration in the global system to be more exposed to WHO recommendations and thus more rapidly to report a first case of COVID-19 and implement the closure of schools. These data have been obtained from Gygli [21]. To facilitate interpretation of this index we transformed values to z-scores.

**Gross domestic product per capita.**   We employ a measure of gross domestic product (GDP) per capita (purchasing power parity for 2000 US$). We log-transformed this variable to avoid influence of outliers because of the skewed distribution. We expect this variable to increase the speed of adoption of reporting the first case since early detection will help them to determine more rapidly what course of actions to follow. We also expect this variable to delay the implementation of school closure since this measure can affect the functioning of the economy by reducing mobilization of their citizens and thus affecting productivity. On the other hand, a poorer nation-state, which lacks financial means, advanced technological measures to accelerate testing, or efficient health systems, may turn to wide school closure simply because this is its most available policy. These data have been obtained from the World Bank [22].

**Population size.**   Countries with larger populations may be more concerned on how to avoid an outbreak crisis in severely impacting the health system, and thus more rapidly testing and finding the presence of the virus in the population. On the other hand, larger populations may require higher levels of internal coordination to effectively close schools and therefore may delay the implementation of this measure. These data have been obtained from the WB [22].

**Democracy index.**   We adopt a measure of democracy, which identifies nations along a scale ranging from 0 ('strongly autocratic') to 100 ('strongly democratic'). The Democracy Index is based on five categories: electoral process and pluralism; civil liberties; the functioning of government; political participation; and political culture. Regimes that enjoy higher levels of democracy may delay the implementation of school closing since this measure contradicts core values and beliefs associated with respecting personal liberties. These data have been obtained from the Economic Intelligence Unit database for the year 2019 [23]. To facilitate interpretation of this index we transformed values to z-scores.

**Economic integration to China.**   This is a measure of how much integrated a country is to China's exports. We measured the total of all products exported value to each country from China divided by GDP. The larger the proportion of this value, the more integrated to China's commerce a country is. This is a proxy to measure economic integration. We chose this

country since the first world case of COVID-19 was thereby reported. Countries more integrated with China are expected to both increase the speed of reporting the first case of COVID-19 as well as introducing school closure to avoid the spreading of the virus. These data have been obtained from United Nations Comtrade database [24].

**Economic integration to Republic of Korea.** This is a measure of how much integrated a country is to Republic of Korea's exports. We measured the total of all products exported value to each country from Republic of Korea divided by GDP. The larger the proportion of this value, the more integrated to Republic of Korea's commerce a country is. We chose this country since it was the second one in reporting a severe outbreak of COVID-19. Countries more integrated with Republic of Korea are expected to both increase the speed of reporting the first case of COVID-19 as well as introducing school closure to avoid the spreading of the virus. These data have been obtained from United Nations Comtrade database [24].

**Economic integration to Italy.** This is a measure of how much integrated a country is to Italy's exports. We measured the total of all products exported value to each country from Italy divided by GDP. The larger the proportion of this value, the more integrated to Italy's commerce a country is. We chose Italy since this was the most shocked country in reporting high levels of lethality at the time WHO declared a global emergency [25]. Countries more integrated with Italy are expected to both increase the speed of reporting the first case of COVID-19 as well as introducing school closure to avoid the spreading of the virus. These data have been obtained from United Nations Comtrade database [24].

For each determinant we used the last year in which countries reported the respective information.

## iii) Methods

To obtain valid estimates to examine policy adoption, we employ survival analysis. This method allows explaining events occurring to countries over a specified period [26]. Survival analysis has been used for various types of events ranging from decolonization [27] to policy adoption [28]. We particularly use the Weibull hazard function since its $\rho$ value can be used to interpret whether policy adoption significantly increases during the observed period. The Weibull function ($h_0(t)$) is specified as $h_0(t) = \rho * t^{\rho - 1}$. If $\rho$ is less than 1, the speed of policy adoption (i.e. hazard of failure) decreases with time, while if it is greater than 1, the speed of the policy adoption increases with time. We hypothesize that, if the thesis of convergence is supported, the $\rho$ value will be greater than 1, because it would run counter to the heterogeneity bias. In reporting the results we call this shape parameter "speed," [28] as its sign and magnitude provide information on whether baseline adoption increases or slows during the observed period. For the thesis of convergence to be supported by the results, the parameter $\rho$ should increase significantly, because it would run counter to the heterogeneity bias. However, a lower parameter $\rho$ can be the product of high-hazard countries, which leave behind the group of low-hazard cases leading to the suggestion that the overall parameter has declined with time. If the convergence process was a response to a national stimulus, with those countries most predisposed to reporting or adopting first, then the parameter would not increase as the first adopters were censored. If instead an ongoing global diffusion process is boosting the adoption of the two events, a significant increase in the parameter of the models should be observed.

It is important to notice that since outcomes could be a result of modeling countries as if they had been equally or not exposed to the same time risk, we defined three different onsets of risk: i) December 31st, 2019, when China alerted WHO's authorities to a cluster of pneumonia in Wuhan; ii) January 31st, 2020, when WHO declared COVID-19 to be a global health

emergency; and iii) the first case detected in each country. The first two were used to assess when the first case of COVID-19 was reported per country, and each onset of risk were used to predict school closure. Information to determine the two first onsets were derived from WHO's press conferences [11, 24].

Since unobserved heterogeneity could also arise from information that countries share due to their regional closeness, implying that unobserved processes could bias the results of the parameters [26], we adjusted the precision of the estimates for their adoption rates in reference to 22 regional clusters based on the United Nations geoscheme [29] (S2 File has the regional cluster list with the countries). In other words, each regional cluster was assigned a random effect—whose distribution does not depend on the observed variables—to model the potential impact of information exchange among countries within each cluster.

When needed, differences in the association of parameters were tested by comparing the value of d/SEd to the standard normal distribution, where d is the difference between the two estimates, and $⟦SE⟧\_d = \sqrt{(⟦SE⟧\_1^2 + ⟦SE⟧\_2^2)}$ is the standard error of the difference [30].

We carried out several sensitivity analyses to (1) indirectly assess whether the results were robust to model specification and (2) using alternative distributions (exponential, and Gompertz models) (Tables S3.1 and S3.2 in S3 File). We also carried out sensitivity analysis with countries in which national decisions were not taken but had begun a process by closing school in states or provinces. In this case we used the date in which the last state or province had close schools (Table S4.1 in S4 File). We also use linear regression and negative binomial models assuming that countries were independent of each other at the time of closing schools (Tables S5.1 and S5.2 in S5 File). We used Stata/SE 14.0 for all the analyses (codes available in S6 File) [31].

## Results

### Global *viralization* of the COVID-19 and school closures

Since December 31[st] 2019, up until June 1[st] 2020, we tracked 194 member states of United Nations, and successfully complied information for both events for 186 countries. This sample corresponds to 99,65% of the world population. In this period, 172 countries had reported the presence of COVID-19 in their territories, and 165 closed their schools at the national level. Fig 1 depicts the cumulative distribution of both events. In that period Australia, Russia, Seychelles, and United States closed schools on a regional or state basis rather than nationally, and Botswana El Salvador, Guinea-Bissau, Kyrgyzstan, Lao People's Democratic Republic, Libya, Malawi, Montenegro, Myanmar, San Marino, Saint Kitts and Nevis, Sao Tome and Principe, Syrian Arab Republic and Yemen close schools before the first case was reported. In Table 2, we depict the average of the number of days since December 31[st] 2019 in which countries reported the first detected case of COVID-19 (mean: 62 days; SD: 18) and when nationally schools closures happened (mean: 77 days; SD 7). The average of number of days in which countries nationally closed their schools, since their first case of COVID-19 was 15 days (SD: 16).

### Global spreading of first reported cases and national characteristics

Table 3 reports the structural parameter ρ in which the first reported cases of COVID-19 are analyzed using two different onset risk. We observe that the structural parameters speed of adoption (ρ) capture increases of 5.85 (95% CI: 4.03, 8.49) for the onset December 31[st], 2019, when China reports to WHO's authorities the epidemic in Wuhan, and 10.34 (95%CI: 8.11, 13.18), for the January 31[st] 2020, when WHO declares global health emergency. Both values indicate that the speed of detecting the first case of COVID-19 is significant growing over

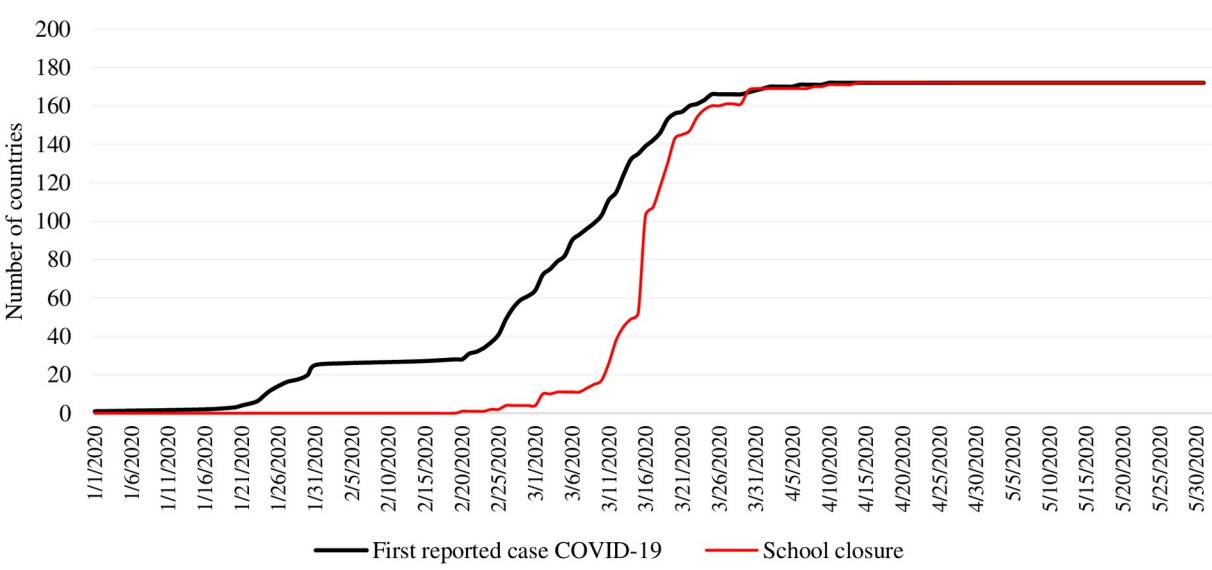

**Fig 1. Daily cumulative number of first reported case COVID-19 and school closure.**

**Table 2. Descriptive statistics of dependent and independent variables, sources and predicted effect.**

| Variables | Mean | SD | Min | Max | Predicted effects |
|---|---|---|---|---|---|
| *Outcomes* | | | | | |
| *Number of days since first case of COVID-19 was reported after December 31st 2019 (China reports to WHO authorities)* [a] | 62 | 18.1 | 16 | 101 | |
| *Number of days since first case of COVID-19 was reported after January 31st 2019 (WHO declares a Global International Emergency)* [a] | 36 | 12.4 | 0 | 70 | |
| *Number of days since school closures were declared after December 31st 2019 (China reports to WHO authorities)* [a] | 77 | 7.2 | 51 | 105 | |
| *Number of days since school closures were declared after January 31st 2019(WHO declares a Global International Emergency)* [a] | 45 | 7.2 | 20 | 74 | |
| *Number of days since school closures were declared after first case was detected* [a] | 15 | 16.6 | 0 | 76 | |
| *Determinants* | | | | | |
| *Epidemic security index (z score)* | 0.0 | 1.0 | -1.84 | 3.47 | Increase in reporting the first COVID-19 case and slower adoption of school closure |
| *Globalization index (z score)* | 0.0 | 1.0 | -1.96 | 1.95 | Increase in reporting the first COVID-19 case and swifter adoption of school closure |
| *GDP per capita (ln)* | 8.7 | 1.5 | 5.6 | 13.6 | Increase in reporting the first COVID-19 case and slower adoption of school closure |
| *Population size (ln)* | 15.8 | 2.0 | 9.8 | 21.1 | Increase in reporting the first COVID-19 case and slower adoption of school closure |
| *Democracy index (z score)* | 0.0 | 1.0 | -2.23 | 1.94 | Increase in reporting the first COVID-19 case and slower adoption of school closure |
| *Economic Integration to China* | 1.2 | 7.3 | 0.0 | 97.4 | Increase in reporting the first COVID-19 case and swifter adoption of school closure |
| *Economic Integration to Republic of Korea* | 10.1 | 116.31 | 0.0 | 1556.7 | Increase in reporting the first COVID-19 case and swifter adoption of school closure |
| *Economic Integration to Italy* | 1.3 | 2.9 | 0.0 | 30.2 | Increase in reporting the first COVID-19 case and swifter adoption of school closure |

[a] List with all sources is available in S1 File.

**Table 3. Weibull models predicting first reported case of COVID-19.**

| Outcome | First reported case of COVID-19 | | | | | | | |
|---|---|---|---|---|---|---|---|---|
| Onset | December 31st, 2019-China reports to WHO's authorities the epidemic in Wuhan | | | | January 31st, 2020-WHO declares global health emergency | | | |
| Determinants | HR | 95% CI | | HR | 95% CI | | HR | 95% CI | |

| Determinants | HR | 95% CI | | HR | 95% CI | | HR | 95% CI | |
|---|---|---|---|---|---|---|---|---|---|
| Epidemic security index (z score) | **1.25** | **1.02** | **1.57** | **1.26** | **1.03** | **1.57** | **1.37** | **1.09** | **1.72** | **1.37** | **1.10** | **1.72** |
| Globalization index (z score) | 1.13 | 0.82 | 1.55 | 1.20 | 0.84 | 1.71 | 1.38 | 0.84 | 2.28 | 1.39 | 0.83 | 2.31 |
| GDP per capita (ln) | **1.63** | **1.30** | **2.03** | **1.54** | **1.20** | **1.98** | **1.74** | **1.29** | **2.35** | **1.74** | **1.29** | **2.35** |
| Population size (ln) | **1.34** | **1.19** | **1.52** | **1.31** | **1.17** | **1.48** | **1.31** | **1.09** | **1.58** | **1.31** | **1.09** | **1.55** |
| Economic Integration to China | 1.14 | 0.96 | 1.34 | | | | 1.05 | 0.88 | 1.25 | | | |
| Economic Integration to Republic of Korea | | | | **1.00** | **1.00** | **1.00** | | | | 1.00 | 0.99 | 1.00 |
| Economic Integration to Italy | 0.99 | 0.99 | 1.00 | 0.99 | 0.99 | 1.00 | 1.00 | 0.99 | 1.00 | 1.00 | 0.99 | 1.00 |
| Speed of adoption ($\rho$) | **5.85** | **4.03** | **8.49** | **6.01** | **4.16** | **8.69** | **10.34** | **8.11** | **13.18** | **10.31** | **8.15** | **13.07** |
| Number of countries | | 164 | | | 165 | | | 141 | | | 143 | |
| Number of adoptions | | 164 | | | 165 | | | 141 | | | 143 | |
| Time at risk | | 10316 | | | 10411 | | | 5351 | | | 5404 | |

All models adjusted for clustering at the region level. CI Confidence Interval. HR Hazard Ratio.

time. For the first onset, countries are 6 times more likely to report first cases after ten days, than after 7 days ($(10/7)^{5.85-1}$). Whereas after the WHO declared the global emergency, countries were more than 28 times more likely to report cases ($(10/7)^{10.34-1}$). In other words, the speed of reporting first cases is faster after the WHO declared a global emergency. In (S8.1 Fig in S8 File) we observe the distribution of both structural parameters from day 1 to day 14, with 7 days as the base to compare.

In terms of whether national characteristics explain a more rapid response to report, we observe that health systems designed to respond and mitigate the spread of an epidemic, GDP per capita and population significantly increased the hazard ratio of this event in both models. After the WHO declared a global emergency, we observe that one standard deviation increase in the epidemic security index, and a 1-log increase in GDP per capita and population were more likely to increase the rate of report first cases by 37% (Hazard Ratio (HR) 1.37 (95% CI: 1.00,1.02)) 74% (HR 1.74 (95% CI:1.29, 2.35)) and 34% (HR 1.34 (95% CI:1.19, 1.52)). Regarding the other three variables only economic integration to Republic of Korea suggests a more rapid response to detect the first case before the WHO declared a global emergency. More specifically, a country is 1% (HR: 1.00 (95%CI: 1.00, 1.00)) more likely to identify a first case the more commercially integrated to Republic of Korea is.

## Global spreading of school closures and national characteristics

In Table 4, we report results regarding school closures as policies adopted to curb down the pandemic at the national level. The structural parameter speed of adoption ($\rho$) captures significant increases of 11.48 (95% CI: 9.67, 13.62), and 8.52 (95%CI: 6.64, 10.94), indicating that regardless of the onset chosen the speed of closing schools has progressively augmented (as Fig 1 depicts). However, opting for different onset of risk changes the global speed of adoption of this policy. Ten days after the WHO declares a global health emergency, countries are 42 times more likely to close schools than after 7 days ($(10/7)^{11.48-1}$), whereas after each country reported its first case, countries were 15 more times likely close schools ($(10/7)^{8.52-1}$). In (S8.2

**Table 4. Weibull models predicting school closure.**

| Outcome | Date in which schools were closed at the national level | | | | | | | | |
|---|---|---|---|---|---|---|---|---|---|
| Onset | December 31st, 2019-China reports to WHO's authorities the epidemic in Wuhan | | | January 31st, 2020-WHO declares global health emergency | | | Respective date a country reports its first case of COVID-19 | | |
| Determinants | HR | 95% CI | | HR | 95% CI | | HR | 95% CI | |
| Epidemic security index (z score) | **0.64** | **0.51** | **0.80** | **0.64** | **0.51** | **0.80** | **0.64** | **0.50** | **0.82** |
| Globalization index (z score) | **2.05** | **1.43** | **2.94** | **2.05** | **1.43** | **2.94** | **1.74** | **1.34** | **2.24** |
| GDP per capita (ln) | 0.96 | 0.72 | 1.26 | 0.96 | 0.73 | 1.26 | 0.92 | 0.70 | 1.21 |
| Population size (ln) | 1.14 | 0.94 | 1.36 | 1.13 | 0.95 | 1.37 | 1.06 | 0.89 | 1.26 |
| Democracy (z score) | 0.71 | 0.44 | 1.14 | 0.71 | 0.44 | 1.14 | 0.70 | 0.44 | 1.11 |
| Economic Integration to Republic of Korea | 0.99 | 0.99 | 1.00 | 0.99 | 0.99 | 1.00 | 0.99 | 0.99 | 1.00 |
| Economic Integration to Italy | **1.00** | **1.00** | **1.00** | **1.00** | **1.00** | **1.00** | 1.00 | 0.99 | 1.00 |
| Speed of adoption ($\rho$) | **11.48** | **9.67** | **13.62** | **11.48** | **9.67** | **13.62** | **8.52** | **6.64** | **10.94** |
| Number of countries | | 143 | | | 143 | | | 128 | |
| Number of adoptions | | 139 | | | 139 | | | 124 | |
| Time at risk | | 10939 | | | 6506 | | | 2133 | |

All models adjusted for clustering at the region level. CI Confidence Interval. HR Hazard Ratio.

Fig in S8 File) we observe the distribution of both structural parameters from day 1 to day 14, with 7 days as the base to compare.

In terms of national characteristics, and regardless of the model, we observe a significant decreased in hazard ratios of relations to the epidemic security index, and concomitant increases in the globalization index. After each country reported a first case, for a one standard deviation increase in the epidemic security index, delays of school closures by 36% (HR 0.64 95% CI: 0.51, 0.80) in standard deviation units are observed. Whereas, for a one standard deviation increase in the globalization index, accelerations of school closures by 74% (HR 1.74 95% CI: 1.34, 2.24) in standard deviation units are noticed. We also observe that, after the WHO declared the international emergency, countries more commercially integrated to Italy were 1% (HR 1.00 (95%CI: 1.00, 1.00) more rapid to adopt this measure. GDP per capita, population, democracy and economic integration to Republic of Korea were not associated with time to the adoption of school closure. Sensitivity analyses showed that results were robust to model specification, and alternative distributions (including countries in which a national decision was not yet taken using the date in which the last state or province implemented school closure (S3.1, S3.2 Tables in S3 File, S4.1 Table in S4 File, S5.1, S5.2 Tables in S5 File).

## Discussion

The viralization of COVID-19 and policies to contain it across the world has been swift. At least 172 countries reported the presence of a case after China informed WHO's authorities of the Wuhan's cluster. From the date in which WHO declared a global emergency, January 31st 2020, up until June 1st 2020, the average of detecting the first case was slightly more than one month. Once countries detected the first case, it took them in average two weeks to close schools at the national level. In this paper, we advanced two groups of hypotheses to understand the speed of this global phenomenon, and our preliminary findings partially confirmed them. After the global declaration of emergency of the WHO, countries were more rapidly to detect a first COVID-19 case. Further, if countries had well designed health systems to respond

to an epidemic, were rich and highly populated were more likely to report a first COVID-19 case in their territories. Countries were also faster in adopting school closure, if they were more globally integrated and had stronger ties to Italy, but slower if their health system was better prepared to contain an epidemic. To sum up an interaction of international with national forces is likely to explain the speed of different layers associated with this phenomenon.

We are witnessing indeed an unprecedent case of global convergence. Previous studies of policy adoption which focused on periods of more than 30 years, using country-year as unit of analysis, have detected increases in the shape speed of adoption parameter ρ above 2 when measuring decision to implement privatization programs [28] or road safety polices [3]. In this study, using country-day as unit of analysis, the most conservative parameter ρ is higher than 5, suggesting a unique case of rapid global convergence. An important finding in this regard is that the speed of detection the first case of COVID-19 was much higher after the global emergency declaration than when China notified to WHO's authorities. After the global emergency declaration countries were significantly more rapid to detect the first case of COVID-19. This suggests that after the WHO scaled its global response, countries were more likely to report the presence of positive cases, and in better condition to initiate the implementation of health and related policies to contain the outbreak. Hence the importance of international organizations with high credibility in times of high uncertainty.

Our results indicate that those with better prepared health systems, richer and more populated were associated with higher hazard rates to inform the presence of first cases. These findings could be signaling more capacity to test and increases in the probability that citizens from these countries were returning from COVID-19 risk zones. To detect whether more populated countries for instance responded more rapidly due to concerns of observing outbreaks, more information is needed to determine how countries were targeting the identification of these cases. Nevertheless, to assess the robustness of the variable Epidemic security index, we also used a variable from the same global index 'early detection and reporting epidemics of potential international concern,' (which systematizes information regarding the quality and presence of laboratory systems, real time surveillance and reporting systems, epidemiology workforce, and data integration between human/animal/environmental health sectors), and results were robust in direction and magnitude when compared to how well prepared the health systems were when targeting an epidemic (results available in S7.1 Table in S7 File).

Our second group of hypotheses of why countries may adopt more rapidly or delay the policy of school closure identified again the importance of global and national factors. First, the presence of well-prepared health systems designed to tackle potential outbreaks was associated with delays in implementing school closures. This could be associated with a better health management and understanding of when the introduction of more stringent measures is necessary. This is particularly salient at the time that knowledge regarding population transmission characteristics was still developing. Second, how well connected the country is to the global system, measured with the globalization index, increases the speed of implementing this policy. A possible explanation of this finding is that more open countries are likely to receive faster information of what measures should be set in place in a global emergency. Further, countries with higher level of integration to the global system are more exposed to influences of international organizations [7, 9], in which the role of WHO, in a global health crisis can be indeed very relevant. It is important to note however that this policy was not a top priority in the set of recommendations diffused by the WHO to contain the spread of COVID-19 [32]. Indeed the WHO's first report regarding China, only recommended the possibility of considering closing schools for countries in which imported cases of COVID-19 had been reported [33], but only after some simulations were carried out. This recommendation could have been

taken by countries which reported the first case, as a positive signal to proceed, even if countries did not have the capacity to develop the recommended simulations. In reference to our finding of integration to Italy, the rapid adoption of school closure of these countries could be explained by the constant information received of the progression of the outbreak in which school closure was one of the measures taken. This resembles a process of policy diffusion whereby policies are triggered by the events happening in other closely related country. As it has been pointed out, this particular process of diffusion emerges when high levels of uncertainty are present [4, 7]. In short, countries with closer economic ties to Italy were more rapid to adopt this measure since this could help them anticipating more successfully how to tackle a likely outbreak. Lastly in terms of economic and political capacities, we observed that none of the variables representing these dimensions at the national level were adequate to predict how fast or slow countries implemented school closures. This ultimately reinforces the importance of global forces and considering adequate health national variables when assessing which responses were applied.

While this study has limitations it also opened new questions to better understand this phenomenon. First, since we only conceptualize countries which nationally determined the closures of schools as adoption cases, other analyses are needed to understand patterns of countries where closure of schools was decided at a subnational level, like United States or the Russian Federation. However, to check for the robustness of our results, we also carry out analyses in which we used the last date in which a state or province closed schools in countries where a national decision was not reached, and results were consistent (see analysis in S4 File). Second, while our preliminary analyses showed specific patterns of globalization to examine some aspects of the implication of COVID-19, we should emphasize the need for greater precision and granularity when examining more critically the processes of policy diffusion of school closures and first case reporting at the global level. Future analyses should attempt to better capture how actors, who transit international and nation social networks, debated, accepted and in some cases rejected the implementation of school closure within the context of other recommended policies but also more controversial ones such as full lockdowns. While findings of the current study highlight the great importance of time in the diffusion of policies, particularly in the context of a pandemic, these results call for a larger expansion of the way we understand the actions of political and economic actors and scientists, in national and international arenas. While the rapid detection of first cases may have triggered national efforts to avoid outbreaks, it is very much unclear the extent under which the rapid or slower adoption of school closure, and other policies, will have an impact in reducing both the spread and lethality of COVID-19. A more sophisticated analysis at the end of this pandemic, with inclusion of data on the effects of school closures and other implemented interventions will inform future policies about timing of implementation of such policies and their efficacy.

## Supporting information

**S1 File. Dates and sources.**
(DOCX)

**S2 File.**
(DOCX)

**S3 File. Analysis with Gompertz and exponential.**
(DOCX)

**S4 File.**
(DOCX)

**S5 File.**
(DOCX)

**S6 File.**
(DOCX)

**S7 File.**
(DOCX)

**S8 File.**
(DOCX)

## Author Contributions

**Conceptualization:** José Ignacio Nazif-Muñoz, Youssef Oulhote.

**Data curation:** José Ignacio Nazif-Muñoz, Youssef Oulhote.

**Formal analysis:** José Ignacio Nazif-Muñoz, Sebastián Peña.

**Funding acquisition:** José Ignacio Nazif-Muñoz.

**Investigation:** José Ignacio Nazif-Muñoz, Sebastián Peña, Youssef Oulhote.

**Methodology:** José Ignacio Nazif-Muñoz.

**Project administration:** José Ignacio Nazif-Muñoz.

**Resources:** José Ignacio Nazif-Muñoz.

**Supervision:** José Ignacio Nazif-Muñoz.

**Validation:** José Ignacio Nazif-Muñoz, Sebastián Peña, Youssef Oulhote.

**Visualization:** José Ignacio Nazif-Muñoz.

**Writing – original draft:** José Ignacio Nazif-Muñoz.

**Writing – review & editing:** José Ignacio Nazif-Muñoz, Sebastián Peña, Youssef Oulhote.

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
