## [Decision Letter · Decision Letter 0]

6 Jan 2021

PONE-D-20-17466

The global viralization of policies to contain the spreading of the COVID-19 pandemic: analyses of school closures and first reported cases.

PLOS ONE

Dear Dr. José Ignacio Nazif-Munoz,

Thank you for submitting your manuscript to PLOS ONE. After careful consideration, we feel that it has merit but does not fully meet PLOS ONE’s publication criteria as it currently stands. Therefore, we invite you to submit a revised version of the manuscript that addresses the points raised during the review process.

Kindly address all the comments by the reviewers.

Check PLoS ONE reference style and adhere.

Explore how else you could make figure 2 more informative.

Be more explicit about your data collection from multiple sources in 172 countries. Are the sources open repositories? if no, how did you obtain permission to extract data from the other closed sources?

We look forward to receiving your revised manuscript.

Kind regards,

Olanrewaju Oladimeji, Ph.D., MB; BS

Academic Editor

PLOS ONE

Additional Editor Comments:

Kindly address all the comments by the reviewers.

Check PLoS ONE reference style and adhere.

Explore how else you could make figure 2 more informative.

Be more explicit about your data collection from multiple sources in 172 countries. Are the sources open repositories? if no, how did you obtain permission to extract data from the other closed sources?

Journal Requirements:

3. We note that Figure 2 in your submission contain map images which may be copyrighted. All PLOS content is published under the Creative Commons Attribution License (CC BY 4.0), which means that the manuscript, images, and Supporting Information files will be freely available online, and any third party is permitted to access, download, copy, distribute, and use these materials in any way, even commercially, with proper attribution. For these reasons, we cannot publish previously copyrighted maps or satellite images created using proprietary data, such as Google software (Google Maps, Street View, and Earth). For more information, see our copyright guidelines: http://journals.plos.org/plosone/s/licenses-and-copyright.

3.1. You may seek permission from the original copyright holder of Figure 2 to publish the content specifically under the CC BY 4.0 license. 

3.2. If you are unable to obtain permission from the original copyright holder to publish these figures under the CC BY 4.0 license or if the copyright holder’s requirements are incompatible with the CC BY 4.0 license, please either i) remove the figure or ii) supply a replacement figure that complies with the CC BY 4.0 license. Please check copyright information on all replacement figures and update the figure caption with source information. If applicable, please specify in the figure caption text when a figure is similar but not identical to the original image and is therefore for illustrative purposes only.

"No"

Reviewers' comments:

Reviewer's Responses to Questions

**Comments to the Author**

1. Is the manuscript technically sound, and do the data support the conclusions?

Reviewer #1: Yes

Reviewer #2: Yes

2. Has the statistical analysis been performed appropriately and rigorously? 

Reviewer #1: Yes

Reviewer #2: Yes

3. Have the authors made all data underlying the findings in their manuscript fully available?

Reviewer #1: Yes

Reviewer #2: Yes

4. Is the manuscript presented in an intelligible fashion and written in standard English?

Reviewer #1: Yes

Reviewer #2: Yes

5. Review Comments to the Author

Reviewer #1: Manuscript number- PONE-D-20-17466

Full title: The global viralization of policies to contain the spreading of the COVID 19 pandemic: analyses of school closure and first reported cases

Abstract: The abstract is clear and concise. It summarises the article adequately to provide the reader with the essence of the paper. Paragraph 3 of page 2 (line 2 and 3): the width of the CI were missing and requires correction.

Introduction: This is a research paper that addresses an interesting and relevant area of research. This section illustrates the context of the research topic. The authors presented their plausible hypothesis well. However, the paragraph listing the hypotheses 1.1 to 2.5 is difficult to read. It may be beneficial to use a figure with the list of the hypotheses.

Methods and data: The authors provided the sources of the data with explanations for the choice where necessary. This section presents an adequate description and explanation of the methods used in the study. The authors provided additional measures to ensure reliability and validity. There was rigour demonstrated in the selection of data sources, data collection and the analysis. The readers are informed of the methods used to check for items like heterogenicity, sensitivity test etc.

Results and discussion: The authors presented the results using figures and tables, which made the reading easy for the reader. The tables were clear and assisting the reader in understanding the numbers. The authors performed statistical analysis appropriately and rigorously.

The assumptions of the results are explained and supported by raw data. The authors provided all the relevant data underlying the findings of the study as annexures. The study appropriately interpreted the results. The discussion is well written, supported by fitting references.

Conclusion: This research paper derived its conclusions from the study findings.

References: The paper cited adequate and appropriate references.

General: The research has been conducted rigorously, with the appropriate data source, data collection, analysis and interpretation. The language used is clear and straightforward and easy to understand.

Reviewer #2: The authors should consider revising the opening statement of the background of the abstract such that "with" will not be used.

The last sentence of the methods section of the abstract appears to be inappropriately placed as it is part of the objective(s) of the study and not methods or approaches to conducting this study.

The results section included "Ten days after the World Health Organization (WHO) declared COVID-19 to be an international emergency" It is important to include the exact date that this declaration was made in the background section of the abstract as stated in the introduction section of the manuscript.

This statement "Ten days after the World Health Organization (WHO) declared COVID-19 to be an international emergency, countries were 28 (95% CI: 12,77) times more likely to report first COVID-19 cases and 42 (95% CI: 22,90) times more likely to close schools" in the results section is unclear as there was no reference for comparison for the measures of effects used, was it as compared 7 days depicted in the results section of the manuscript?

The authors may need to include measures taken to verify and or validate the dates in the outcomes sub-section of the methods and data section of the manuscript even if appendix 1 contains all sources per country.

The authors should provide information on the rationale for compiling and analyzing data on 172 countries only. Does it mean that these 172 countries were the only countries who implemented national decision of school closure before 1st of May 2020.

6. PLOS authors have the option to publish the peer review history of their article (what does this mean?). If published, this will include your full peer review and any attached files.

Reviewer #1: No

Reviewer #2: **Yes: **Tolulope Olumide Afolaranmi

---

## [Author Response · Author response to Decision Letter 0]

11 Jan 2021

We have uploaded a cover letter with responses to reviewers and the editor

---

## [Decision Letter · Decision Letter 1]

8 Mar 2021

The global viralization of policies to contain the spreading of the COVID-19 pandemic: analyses of school closures and first reported cases.

PONE-D-20-17466R1

Dear Dr. Nazif-Munoz,

We’re pleased to inform you that your manuscript has been judged scientifically suitable for publication and will be formally accepted for publication once it meets all outstanding technical requirements. In this regard, the author(s) should revise the abstract, whereas references to the empirical outcomes are required.

Kind regards,

Stefan Cristian Gherghina, PhD. Habil.

Academic Editor

PLOS ONE

Additional Editor Comments (optional):

Reviewers' comments:

Reviewer's Responses to Questions

**Comments to the Author**

1. If the authors have adequately addressed your comments raised in a previous round of review and you feel that this manuscript is now acceptable for publication, you may indicate that here to bypass the “Comments to the Author” section, enter your conflict of interest statement in the “Confidential to Editor” section, and submit your "Accept" recommendation.

Reviewer #1: All comments have been addressed

Reviewer #2: All comments have been addressed

2. Is the manuscript technically sound, and do the data support the conclusions?

Reviewer #1: Yes

Reviewer #2: Yes

3. Has the statistical analysis been performed appropriately and rigorously? 

Reviewer #1: Yes

Reviewer #2: Yes

4. Have the authors made all data underlying the findings in their manuscript fully available?

Reviewer #1: Yes

Reviewer #2: Yes

5. Is the manuscript presented in an intelligible fashion and written in standard English?

Reviewer #1: Yes

Reviewer #2: Yes

6. Review Comments to the Author

Reviewer #1: Recommendations were made on the first review and the authors have attended to the concerns raised in the first review satisfactorarily.

Reviewer #2: The authors have addressed significantly the review comments. However, there are a few minor revisions required as outlined below;

Abstract: The authors should ensure that the background to the abstract is an excerpt of the content of the introduction of the manuscript.

Discussion: Making reference to results in the discussion section such as "results

available in Supporting information S7 Table S7.1." First paragraph pages 20 and First paragraph page 22,

7. PLOS authors have the option to publish the peer review history of their article (what does this mean?). If published, this will include your full peer review and any attached files.

Reviewer #1: No

Reviewer #2: **Yes: **Tolulope Olumide Afolaranmi

---

## [Editor Report · Acceptance letter]

10 Mar 2021

PONE-D-20-17466R1 

The global *viralization* of policies to contain the spreading of the COVID-19 pandemic: analyses of school closures and first reported cases. 

Dear Dr. Nazif-Muñoz:

I'm pleased to inform you that your manuscript has been deemed suitable for publication in PLOS ONE. Congratulations! Your manuscript is now with our production department. 

Kind regards, 

on behalf of

Dr. Stefan Cristian Gherghina 

Academic Editor

PLOS ONE